# Influence of Parental Perception of Child’s Physical Fitness on Body Image Satisfaction in Spanish Preschool Children

**DOI:** 10.3390/ijerph20085534

**Published:** 2023-04-17

**Authors:** Jorge Rojo-Ramos, María Mendoza-Muñoz, Antonio Castillo-Paredes, Carmen Galán-Arroyo

**Affiliations:** 1Physical Activity for Education, Performance and Health, Faculty of Sport Sciences, University of Extremadura, 10003 Caceres, Spain; jorgerr@unex.es; 2Physical and Health Literacy and Health-Related Quality of Life (PHYQoL), Faculty of Sport Science, University of Extremadura, 10003 Caceres, Spain; mamendozam@unex.es (M.M.-M.); mamengalana@unex.es (C.G.-A.); 3Departamento de Desporto e Saúde, Escola de Saúde e Desenvolvimento Humano, Universidade de Évora, Largo dos Colegiais 2, 7000-645 Évora, Portugal; 4Grupo AFySE, Investigación en Actividad Física y Salud Escolar, Escuela de Pedagogía en Educación Física, Facultad de Educación, Universidad de Las Américas, Santiago 8370040, Chile

**Keywords:** physical fitness, self-concept, body image perception, body dissatisfaction, children

## Abstract

It is well known that poor physical fitness is an exponential risk factor in the increase in chronic diseases, not only physical but also psychological. Even in childhood, a critical period of development, the perception of physical fitness plays a fundamental role in the individual’s self-concept of body image. Aim: To find out how self-perceived physical fitness influences self-perceived body image in preschoolers. Methods: 475 preschool pupils were recruited in the schools of Extremadura (Spain). They were administered a sociodemographic questionnaire, the Preschool Physical Fitness Index (IFIS) and the Preschool Body Scale (PBS). Findings: Significant correlations (*p* < 0.05) were observed between body dissatisfaction and perceived physical fitness (IFIS), being higher in girls. In terms of variables, general fitness (<0.001), cardio-respiratory fitness (<0.001), muscular strength (<0.001), speed/agility (<0.001) and balance (<0.001) have a negative, medium and significant association with body dissatisfaction in girls; however, this association was lower in the case of boys. Conclusions: The influence of physical fitness had a clear impact on self-perceived body image. With better findings on self-perceived physical fitness variables (IFIS) there was less body dissatisfaction (PBS), especially in the female sex. The results also showed that parents who perceived their children to be in poorer physical condition had higher body dissatisfaction. Therefore, it would be interesting for the context involved, particularly parents, to implement strategies to improve positive body image through the promotion of physical education and physical fitness at an early age.

## 1. Introduction

Early childhood is a critical time for children’s growth and development, when they begin to become aware of their own body, physical shape and self-image [1].

Physical fitness is important for children as it helps them to maintain a healthy body weight [2], build strong bones and muscles, and improve overall health. On the other hand, a positive body image is also significant for children as it can influence their self-esteem and mental well-being [3].

Physical fitness refers to the body’s ability to perform various physical tasks and activities, such as general condition, endurance, strength, flexibility and balance [4].

Body image concerns a person’s perception of their physical appearance and how they feel about their body [5]. In children, it can be referred to as an “inner mirror” or a “picture of our body” [6].

The assessment of physical fitness in children is of great public health relevance, as indicators of physical fitness in childhood have been found to trigger chronic diseases in adulthood [7]. It is also of great educational importance [8,9,10] and clinical significance [11]. However, fitness testing is not always feasible in studies with large populations where resources are limited. In this sense, the International Fitness Scale (IFIS) allows its measurement in an indirect way. It is a short and simple scale of self-perception of physical fitness, which has a high level of validity, internal consistency and reliability in different European countries and in the Spanish population [12]. The validity and test–retest reliability of the IFIS parent scale are moderately acceptable for assessing physical fitness in children aged 3–5 years [13].

In relation to the self-perception of body image in children, it is also relevant to be able to assess it, as studies have found that children begin to perceive their body image as early as age 3, and that is also when they begin to know how they feel [14]. In addition, the influence of the thin girl stereotype on girls from infancy onwards is underestimated [15]. From that age onwards they already perceive fat as something negative, discriminating the slim figure as positive [16]. Being able to assess body self-perception could help to prevent physical and mental illnesses that emerge in adulthood [7]. One of the scales to assess body dissatisfaction in this age group, Body Scale for Preschoolers [17], is a simple scale in which the child is shown several images and chooses which one most resembles his or her real body and which one most resembles his or her ideal body. It is understandable and adapted to their age. It does not require reading comprehension. In addition, this questionnaire has good validity and moderate to high reliability [17].

Physical fitness can have a positive influence on body image perception [18]. Regular exercise and physical activity can lead to improved physical health and fitness [19], which can in turn lead to increased self-confidence and a more positive body image [20]. Additionally, the process of setting and working towards fitness goals can help individuals develop a more positive self-image and self-esteem [21]. However, it’s also important to note that body image perception is a complex and multi-faceted issue that can also be influenced by a variety of other factors such as societal and cultural pressure [22], self-esteem and mental health.

Therefore, it would be interesting to know the relationship between physical fitness and self-perception of body image, from an early age, when the child is still little contaminated by external information and socially marked stereotypes, and to work on them for the optimal development of the person and the improvement of overall health.

In this sense, the aim of the study is to find out how the perception of physical fitness influences the self-perception of body image in Spanish preschool children.

## 2. Materials and Methods

### 2.1. Participants

A convenience sampling method was used [23] to select the sample. A total of 488 participants were contacted, of which 10 did not provide informed consent, and 3 were deleted because the data were found to be incomplete. Thus, the sample size was 475 pupils (252 boys and 223 girls) in the second cycle of pre-school education (3 to 6 years old) from public schools in the region of Extremadura (Spain). Thirty per cent of the sample (n = 144) were in the first year of infant education, 28.8% (n = 137) were in the second year and 40.8% (n = 194) were in the third year. The mean age was 4.11 years (SD = 0.83). Participants were selected using a convenience sampling method. The inclusion criteria for participants were that the parents or guardians of the students agreed to informed consent and collaboration in the study. This study was conducted in accordance with the Declaration of Helsinki and was approved by the bioethics and biosafety committee of the University of Extremadura (71/2022).

### 2.2. Procedure

In order to access the sample, the collaboration of public educational centres of Early Childhood Education in Extremadura was required. For this purpose, the directory of public centres in Extremadura belonging to the Regional Ministry of Education and Employment was accessed and the e-mail addresses of the three hundred and fifty-nine centres that taught Early Childhood Education were selected. Then, an e-mail was sent to Early Childhood Education teachers informing them about the purpose of the study and inviting them to collaborate in the study. The ten schools that agreed to participate in the study were sent a physical copy of the instrument and the informed consent forms.

An appointment was then arranged at each school where a member of the research team came to administer the instruments to the students. Since the first part of the instrument had to be completed by the parents or guardians of the students, it was essential that each student came to school on the agreed day with part A (IFIS scale) completed correctly. Afterwards, each pupil had to fill in part B (Body Scale for pre-school children). For this, the teacher and the member of the research team read and explained the instrument aloud to ensure that the pupils understood what they had to do. First, they were asked to select the image that most resembled them (perceived figure). Secondly, when all students had completed the previous task, they were asked to select the figure they would like to resemble (desired figure). The average time for students to complete the questionnaire was 5 min.

### 2.3. Instruments

The instrument consisted of a socio-demographic questionnaire, the Preschoolers’ Perception of Physical Fitness Questionnaire (IFIS) and the Body Scale for Preschoolers (Figure 1).

Socio-demographic questionnaire: A questionnaire with three questions regarding gender, age and grade was included.

Body Scale for Preschoolers (BSP): To assess body dissatisfaction in preschoolers, the Body Scale for Preschoolers [17] was used. This instrument uses a male and a female version. Each is composed of two scales. The first one presents four body figures in frontal position and the second one in lateral position. All figures had the same height and were ordered from a low body mass index (weight in kilograms/(height in metres) (assigned value 1) to a high body mass index (assigned value 4) (Figure 1). The first image (1) corresponds to an underweight child (BMI—boy = 13.13; BMI—girl = 13.03), the next (2) a normal weight child (BMI—boy = 16; BMI—girl = 15.06), the third (3), an overweight child (BMI—boy = 17.1; BMI—girl = 17.06), and finally (4), an obese child (BMI—boy = 21.03; BMI—girl = 21.25). Using these scales, the perceived figure and the desired figure were to be determined. The value of the perceived figure was obtained by asking the students “Which figure looks most like you” and the value of the desired figure was obtained by asking the students “Which figure would you like to look like? The following formula was used to calculate body dissatisfaction (BD): BD = desired figure—perceived figure. The authors reported a Fleiss kappa (0.61) using the judgement of ten expert child doctors and a test–retest with preschool children (ρfrontal = 0.40; ρlateral = 0.55).

**Figure 1 ijerph-20-05534-f001:**
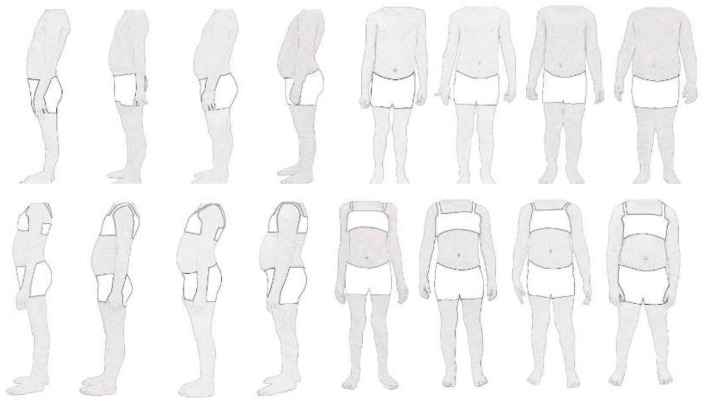
Images corresponding to the male and female sex respectively. Note. From “Development and validation of the preschoolers body scale” [17].

International Fitness Scale (IFIS): this instrument assesses physical fitness perception and was designed and validated by the Promoting Fitness and Health through Physical Activity research group (profith.ugr.es/IFIS) of the University of Granada. For this purpose, an adaptation of the instrument was made, which sought to assess parents’ perception of their child’s physical fitness (compared to their friends). The validity and reliability of the IFIS parent scale for assessing physical fitness in children aged 3 to 5 years has already been shown in a previous study [13]. For them, the statement of the instruction was modified accordingly: “Please think about your fitness level (compared to your friends) and choose the most suitable option” was replaced by “Please think about your child’s level of physical fitness (compared to his/her friends) and choose the most appropriate option”. A Likert scale (1–5) was used, with 1 being very bad, 2 bad, 3 acceptable, 4 good and 5 very good, for the five items (general fitness, cardio-respiratory fitness, muscular strength, speed/agility and balance).

### 2.4. Statistical Analysis

The distribution of the data was explored to assess whether the assumption of normality was met using the Kolmogorov–Smirnov test. As a result, it was found that this assumption was not met, so it was decided to use Spearman’s rho test to analyse the relationship between the body scale for preschool children and the scale for the perception of physical condition (IFIS). To interpret the degree of correlation between the two scales and their dimensions, the ranges proposed by Mondragón-Barrera were taken as a reference: from 0.01 to 0.10 (low correlation), from 0.11 to 0.50 (medium correlation), from 0.51 to 0.75 (considerable correlation), from 0.76 to 0.90 (very high correlation) and from 0.91 to 1.00 (perfect correlation). Cronbach’s alpha was used to analyse the reliability for each of the scales. For the interpretation of the reliability values, the values proposed by Nunally Bernstein were used as a reference: <0.70 (low), 0.71 to 0.90 (satisfactory) and > 0.91 (excellent). All collected data were processed using Statistical Package of Social Science statistical software (version 27, 2021; IBM Corp., IBM SPSS Statistics for MAC OS, Armonk, NY, USA).

## 3. Results

Table 1 presents the results of the correlations between body dissatisfaction (front scale, side scale and the mean value of both scales) and perceived physical condition as a function of gender. It was observed that there were considerable (>0.51) and significant (*p* < 0.05) correlations between body dissatisfaction (front scale, side scale and mean value) and perceived physical fitness (IFIS) in girls, while there were medium (>0.11 and <0.50) and significant (*p* < 0.05) correlations between body dissatisfaction (front, side view and mean value) and perceived physical fitness in boys. To assess the relationship between the body dissatisfaction scale and IFIS, Spearman’s rho test was used.

Table 2 presents the associations between each of the physical fitness variables studied (general fitness, cardio-respiratory fitness, muscular strength, speed/agility and balance) and the body dissatisfaction scales (front, profile and total) as a function of gender. The most remarkable results showed that the parent’s perception on general fitness, cardio-respiratory fitness, muscle strength, speed/agility and balance have a negative, average and significant association with body dissatisfaction in girls. That is, the higher the value on each of the IFIS variables, the lower the body dissatisfaction in girls. However, this association was lower in the case of boys, with lower values than in the case of girls. This association was negative, medium and significant between muscle strength, speed/agility and balance and body dissatisfaction.

Finally, satisfactory reliability values were obtained according to Nunally and Bernstein. Table 3 shows the reliability results for each of the scales used.

## 4. Discussion

The aim of this study was to find out the relationship between physical fitness and perceived body image in children under six years of age, who are in a critical period of development and in which it is crucial to work on these variables in a positive way.

The first finding is that there are considerable and significant inverse correlations between perceived physical condition (IFIS) and body dissatisfaction (BD) (frontal, lateral and mean value) in Spanish preschoolers. This means that the higher the value of the IFIS scale, the lower the level of body dissatisfaction or, in other words, the better the physical fitness, the lower the body dissatisfaction.

Regarding gender, a significant correlation was found in both sexes, being of higher magnitude in girls than in boys, as in a recent systematic review of 16 articles [24], and although this review was not only of children but also of adolescents up to 19 years of age, the findings were very consistent. This may be due to the stereotypical pattern of beauty assumed by girls in the real world [25]. Girls feel more pressure than boys to be stereotypically beautiful as they get older [26]. A recent meta-analysis shows that girls tend to place more importance on their bodies than boys [27]. Other authors agree with our findings, but in a different population, in teachers instead of children [28].

The second finding refers to each of the variables that make up the IFIS. Various meta-analytic reviews have concluded that physical activity is positively related to body image [19,29,30,31]. Correlational data found in the scientific literature have shown a positive relationship between PA, physical fitness and body image in various samples [32]. Experimental research has further shown a positive relationship, such that physically active people have better fitness levels and have a healthier body image [33]. Thus, they have more body satisfaction or, in other words, less body dissatisfaction compared to those who do not engage in PA and their physical condition is lower [31]. The scientific evidence indicates that people with higher levels of PFP show lower body dissatisfaction [34,35]. Furthermore, the majority of students who have a low level of physical activity, are more dissatisfied with their body and more concerned with their body image are those who identify with larger silhouettes [36].

The most remarkable results showed that general fitness, cardio-respiratory fitness, muscular strength, speed/agility and balance have a negative, average and significant association with body dissatisfaction in girls. That is, the higher the value on the scales of each of the IFIS variables, the lower the body dissatisfaction in girls. This is also consistent with other work showing that women tend to be more dissatisfied with their bodies [37]. However, this association was lower in boys, with lower values than in girls [25]. This association was negative, medium and significant between muscle strength, speed/agility and balance and body dissatisfaction, which is consistent with the results of the study, as men seem to attach less importance to their appearance [38].

It is difficult to find results between each of the IFIS dimensions and body dissatisfaction, especially at such young ages. However, there are studies that show that cardio-respiratory training has a positive relationship with body image [39], as well as strength training, which is also positively associated [40,41]. What is clear is that what we do in childhood leaves its mark in adulthood, as the more physically active a person is, the better their self-perception of physical fitness, self-image, functionality and quality of life compared to physically inactive and sedentary people [42].

A relevant aspect to note is that the results of perceived physical fitness were reported by the parents of the participants, based on a previous study [13], which mentioned that the reliability of the parent-reported IFIS is acceptable, but the agreement between parents’ responses and objective measures of fitness levels was poor, suggesting that parents’ responses may not correctly classify preschool children according to their level of physical fitness. This is a limitation of this study, and the results should be taken with caution.

However, the acceptable reliability of the IFIS reported by Sánchez-López et al. [13], as well as that obtained in our study (Cronbach’s alpha = 0.76), highlights another finding of our study. In this sense, the results showed an inverse relationship between body dissatisfaction and parents’ perception of their children’s physical condition, i.e., those parents who perceived their children to be in worse physical condition, their children had greater body dissatisfaction, and this relationship was stronger in the case of girls.

In this sense, there are many studies that show the influence that parents’ perceptions or attitudes can have on their children’s physical activity [43,44,45,46]. Therefore, this information may be very relevant, since the detection of poor parental perceptions of their children’s physical fitness (and consequently, as this study has shown, the relationship of this with their children’s body dissatisfaction), may lead parents to promote healthy habits. That is, making parents aware of their children’s physical condition could lead them to promote healthy habits from an early age, thus influencing balanced nutrition, regular physical activity or limiting screen time, and thus helping to prevent childhood obesity or other pathologies related to sedentary lifestyles.

Therefore, a positive parental perception of their children’s physical condition may contribute to the better emotional and physical well-being of children, and may decrease their body dissatisfaction. However, a negative or overly critical perception may have a negative effect on children’s self-esteem and motivation to be physically active, leading them to be more dissatisfied with their bodies. Ultimately, it is important for parents to be aware of their perception and how it may affect their children, and to foster a positive and encouraging attitude towards physical activity.

### 4.1. Limitations

This work has various limitations. First, due to the convenience sampling used, the findings should be interpreted with care. The socio-demographic data collected is too limited to take into account all the variables that affect the perception of physical fitness and perceived self-image.

As suggested above, a study of the IFIS scale in early childhood report poor criterion validity, suggesting that parents’ responses may not be able to correctly classify preschool children according to their level of physical fitness [13]. Therefore, new studies are needed to rebalance the scale in order to measure the reality of physical fitness, which is so homogeneous in this population that it is difficult for parents to discern.

### 4.2. Implications

It is necessary to create lines of action or implement educational programmes with the aim of improving physical fitness and body image at an early age.

Interdisciplinary work between the administration (which controls the media), the educational community (teachers, parents, carers, pupils) and society in general is of vital importance for the optimal and healthy development of the child.

Parents and caregivers can support children in developing a positive body image [47] by promoting healthy habits, such as regular exercise and healthy eating, and by helping children to understand that everyone’s body is different and that there is no “perfect” body type. Additionally, avoiding negative comments about weight or appearance and avoiding comparisons to other children can help children to develop a positive body image.

## 5. Conclusions

The influence of physical fitness had a clear impact on self-perceived body image. With better findings on self-perceived physical fitness variables (IFIS), there was less body dissatisfaction (PBS), especially in females. The higher the level of physical fitness, the better the self-perceived body image satisfaction, especially in girls. In addition, the results showed that parents who perceived their children to be in poorer physical condition had higher body dissatisfaction, with this relationship being stronger for girls.

Therefore, it is particularly important to monitor both physical fitness perception and body dissatisfaction, with parents playing a key role in fostering a positive and encouraging attitude towards physical activity. It is also important for the public organizations involved to apply strategies and actions to promote physical education and physical activity at an early age, in favour of good physical condition, improving the perception of positive self-image and thus avoiding possible risks of diseases in adulthood.

## Figures and Tables

**Table 1 ijerph-20-05534-t001:** Correlation between body dissatisfaction and International Fitness Scale (IFIS).

Dimensions	IFIS *ρ (p)*	IFIS *ρ (p)*	
		Boy	Girl
(1) Body Dissatisfaction (Front view)	−0.348 (<0.001)	−0.149 (0.018)	−0.528 (<0.001)
(2) Body Dissatisfaction (Side view)	−0.345 (<0.001)	−0.162 (0.010)	−0.532 (<0.001)
(3) Body Dissatisfaction (Mean value)	−0.334 (<0.001)	−0.138 (0.028)	−0.523 (<0.001)

**Table 2 ijerph-20-05534-t002:** Correlation between body dissatisfaction and perceived physical ability.

Variable	General Physical Fitness *ρ (p)*		Cardio-Respiratory Fitness *ρ (p)*		Muscle Strength *ρ (p)*		Speed/Agility *ρ (p)*		Balance *ρ (p)*	
	Boy	Girl	Boy	Girl	Boy	Girl	Boy	Girl	Boy	Girl
(1) Body Dissatisfaction (Front view)	−0.081 (0.199)	−0.371 (<0.001)	−0.089 (0.159)	−0.424 (<0.001)	−0.253 (<0.001)	−0.479 (<0.001)	−0.133 (0.035)	−0.236 (<0.001)	0.148 (0.018)	−0.252 (<0.001)
(2) Body Dissatisfaction (Side view)	−0.116 (0.067)	−0.385 (<0.001)	−0.094 (0.136)	−0.425 (<0.001)	−0.170 (0.007)	−0.431 (<0.001)	−0.170 (0.006)	−0.246 (<0.001)	0.063 (0.317)	−0.311 (<0.001)
(3) Body Dissatisfaction (Mean value)	−0.075 (0.238)	−0.371 (<0.001)	−0.088 (0.162)	−0.438 (<0.001)	−0.216 (0.001)	−0.463 (<0.001)	−0.141 (0.026)	−0.227 (0.001)	0.127 (0.045)	−0.269 (<0.001)

**Table 3 ijerph-20-05534-t003:** Cronbach Alpha values for each scale of the instrument.

Cronbach Alpha	IFIS *ρ (p)*
Perceived figure scale	0.86
Desired figure scale	0.74
IFIS	0.76

## Data Availability

The data presented in this study are available on request from the corresponding author.

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
