# Peer review of "Influence of Parental Perception of Child’s Physical Fitness on Body Image Satisfaction in Spanish Preschool Children"

_ijerph, 2023, doi:10.3390/ijerph20085534_

Round 1
Reviewer 1 Report
This manuscript provides results on a cross-sectional study of Spanish Preschool aged children's Perceived Body Image and their parents' report of their physical fitness. Preschool Physical Fitness In-dex (IFIS) was used to measure the child's physical fitness and the Preschool Body Scale (PBS) was used to measure body image. Body image and the assessment of body satisfaction was based on reports by the preschool child choosing the image that they felt looked like them and the image the would like to be. Physical Fitness was reported by the preschool aged child's parent. Parent reported what they perceived their child's perception as their physical fitness satisfaction. The authors report that the scale used the IFIS had poor criterion validity. I am most concerned that this study more shows how parent perception of child’s fitness impacts the child’s body image. Which is not the conclusion presented by the authors. It would have helped to have a conceptual framework or theory describing how the authors feel this association works. Developmentally preschool aged children's attitudes, preferences and behaviors are most influenced by their families. Overall the manuscript describes the methods well and the analysis is appropriate. However, I feel that without a theory driving this analysis and recognition that the results may more reflect the relationship between the parents' perceptions and attitudes and the impact of that perception on the child's body image. Body dissatisfaction is already impacted their perceived peer and/or parental criticism and influence. I think more research is needed to describe what IFIS represents. It could be a good tool to use for parent counseling related promoting child health and mental health. The authors do not discuss how they deal with children who choose an image for themselves that was not concordant with their gender assigned at birth. How often did this happen?Author Response
REVIEWER 1
Authors’ response: Thank you for your review of our manuscript. We have carefully considered your comments and believe that the quality of the paper has improved after incorporating your suggestions. Below are our responses to your suggestions:
Comments and Suggestions for Authors
This manuscript provides results on a cross-sectional study of Spanish Preschool aged children's Perceived Body Image and their parents' report of their physical fitness. Preschool Physical Fitness In-dex (IFIS) was used to measure the child's physical fitness and the Preschool Body Scale (PBS) was used to measure body image. Body image and the assessment of body satisfaction was based on reports by the preschool child choosing the image that they felt looked like them and the image the would like to be. Physical Fitness was reported by the preschool aged child's parent. Parent reported what they perceived their child's perception as their physical fitness satisfaction. The authors report that the scale used the IFIS had poor criterion validity. I am most concerned that this study more shows how parent perception of child’s fitness impacts the child’s body image. Which is not the conclusion presented by the authors. It would have helped to have a conceptual framework or theory describing how the authors feel this association works. Developmentally preschool aged children's attitudes, preferences and behaviors are most influenced by their families. Overall the manuscript describes the methods well and the analysis is appropriate. However, I feel that without a theory driving this analysis and recognition that the results may more reflect the relationship between the parents' perceptions and attitudes and the impact of that perception on the child's body image. Body dissatisfaction is already impacted their perceived peer and/or parental criticism and influence. I think more research is needed to describe what IFIS represents. It could be a good tool to use for parent counseling related promoting child health and mental health.
Authors’ response: Thank you for your comment, we agree with you and have therefore proceeded to address the results from the perspective you mention, expanding on the discussion and conclusion. Please let us know if you feel that more information should be added and we will do so.
The authors do not discuss how they deal with children who choose an image for themselves that was not concordant with their gender assigned at birth. How often did this happen?
Authors’ response: thank you for your comment, in this case, there were no participants who marked the images corresponding to the opposite sex to yours. That is why no information has been included. If you still consider that some information should be added, please specify and we will do so.
Reviewer 2 Report
This study on self-perceived physical fitness influences self-perceived body image in preschoolers. Even though a convenience sample was taken, it would be informative to have some more information on response rates: from the number of schools approached, how many agreed to participate? And from the students/parents approached, how many agreed to participate and how many of those eventually showed up on the agreed upon day?
The validation procedure of the IFIS physical fitness perception questionnaire is not clear, and no results are given on the outcome of this procedure. It appears that questions are worded as "my condition is.." whereas the parents had to indicate this not about themselves but about their child.
For the instruments it would be good to include example figure of the images presented to the pre-schoolers. It is not clear whether the body figures were based on photographs or drawings/animations. To what extent was gender of the figures visiable? Were faces include or just the torso?
Author Response
REVIEWER 2
Authors’ response: Thank you for your review of our manuscript. We have carefully considered your comments and believe that the quality of the paper has improved after incorporating your suggestions. Below are our responses to your suggestions:
This study on self-perceived physical fitness influences self-perceived body image in preschoolers. Even though a convenience sample was taken, it would be informative to have some more information on response rates: from the number of schools approached, how many agreed to participate? And from the students/parents approached, how many agreed to participate and how many of those eventually showed up on the agreed upon day?
Authors’ response: this information has been added as a suggestion. If you feel that more specific information should be added, please let us know and we will do so.
The validation procedure of the IFIS physical fitness perception questionnaire is not clear, and no results are given on the outcome of this procedure. It appears that questions are worded as "my condition is." whereas the parents had to indicate this not about themselves but about their child.
Authors’ response: Thank you for your comment, to clarify the confusion we have modified the description of the IFIS instrument and the adaptation that was made to it. If you still consider that any further information needs to be clarified please let us know.
For the instruments it would be good to include example figure of the images presented to the pre-schoolers. It is not clear whether the body figures were based on photographs or drawings/animations. To what extent was gender of the figures visiable? Were faces include or just the torso?
Authors’ response: Thank you for your comment, to clarify this aspect we have included the images that were shown to the children in the instrument description.
Round 2
Reviewer 1 Report
I appreciate the authors work to incorporate my comments into the manuscript. I think it is now worded to describe the influence of parental perception of child’s physical fitness on the child’s body satisfaction. A very important revision.
I think the title needs to reflect this results. “Influence of Parental Perception of Child’s Physical Fitness on Body Image Satisfaction in Spanish Preschool Children”
Also check some of the edits to make sure the sentences still make sense. For example in the discussion the words ‘public organisms’ may be better stated as ‘public organizations’.
Author Response
REVIEWER 1
Comments and Suggestions for Authors
I appreciate the authors work to incorporate my comments into the manuscript. I think it is now worded to describe the influence of parental perception of child’s physical fitness on the child’s body satisfaction. A very important revision.
Authors’ response: Thank you very much for your appreciation. Sincerely, thanks to you the quality of the manuscript has improved and becomes more meaningful.
I think the title needs to reflect this results. “Influence of Parental Perception of Child’s Physical Fitness on Body Image Satisfaction in Spanish Preschool Children”
Authors’ response: You are absolutely right. The title is a true reflection of the content of the manuscript. Thank you very much.
Also check some of the edits to make sure the sentences still make sense. For example in the discussion the words ‘public organisms’ may be better stated as ‘public organizations’.
Authors’ response: Thank you for your suggestion. We have modified the sentences to make more sense.